# Comparative Molecular Analysis of Primary Central Nervous System Lymphomas and Matched Vitreoretinal Lymphomas by Vitreous Liquid Biopsy

**DOI:** 10.3390/ijms22189992

**Published:** 2021-09-16

**Authors:** Daniel A. Balikov, Kevin Hu, Chia-Jen Liu, Bryan L. Betz, Arul M. Chinnaiyan, Laxmi V. Devisetty, Sriram Venneti, Scott A. Tomlins, Andi K. Cani, Rajesh C. Rao

**Affiliations:** 1W.K. Kellogg Eye Center, Department of Ophthalmology and Visual Science, University of Michigan, Ann Arbor, MI 48109, USA; dbalikov@med.umich.edu (D.A.B.); laxmiatkuru@gmail.com (L.V.D.); 2Center of Computational Medicine and Bioinformatics, University of Michigan, Ann Arbor, MI 48109, USA; kevhu@med.umich.edu; 3Department of Pathology, University of Michigan, Ann Arbor, MI 48109, USA; liuchiaj@umich.edu (C.-J.L.); bbetz@med.umich.edu (B.L.B.); arul@med.umich.edu (A.M.C.); svenneti@med.umich.edu (S.V.); scott@strataoncology.com (S.A.T.); 4Michigan Center for Translational Pathology, University of Michigan, Ann Arbor, MI 48109, USA; 5Rogel Cancer Center, University of Michigan, Ann Arbor, MI 48109, USA; 6Hematology/Oncology Division, Department of Internal Medicine, University of Michigan, Ann Arbor, MI 48109, USA; 7Center for RNA Biomedicine, University of Michigan, Ann Arbor, MI 48109, USA; 8Division of Ophthalmology, Surgical Service, Veterans Administration Ann Arbor Healthcare System, Ann Arbor, MI 48109, USA; 9A. Alfred Taubman Medical Research Institute, University of Michigan, Ann Arbor, MI 48109, USA

**Keywords:** vitreoretinal lymphoma, primary central nervous system lymphoma, next generation sequencing, precision oncology, precision diagnostics, intratumor heterogeneity

## Abstract

Primary Central Nervous System Lymphoma (PCNSL) is a lymphoid malignancy of the brain that occurs in ~1500 patients per year in the US. PCNSL can spread to the vitreous and retina, where it is known as vitreoretinal lymphoma (VRL). While confirmatory testing for diagnosis is dependent on invasive brain tissue or cerebrospinal fluid sampling, the ability to access the vitreous as a proximal biofluid for liquid biopsy to diagnose PCNSL is an attractive prospect given ease of access and minimization of risks and complications from other biopsy strategies. However, the extent to which VRL, previously considered genetically identical to PCNSL, resembles PCNSL in the same individual with respect to genetic alterations, diagnostic strategies, and precision-medicine based approaches has hitherto not been explored. Furthermore, the degree of intra-patient tumor genomic heterogeneity between the brain and vitreous sites has not been studied. In this work, we report on targeted DNA next-generation sequencing (NGS) of matched brain and vitreous samples in two patients who each harbored VRL and PCSNL. Our strategy showed enhanced sensitivity for molecular diagnosis confirmation over current clinically used vitreous liquid biopsy methods. We observed a clonal relationship between the eye and brain samples in both patients, which carried clonal *CDKN2A* deep deletions, a highly recurrent alteration in VRL patients, as well as *MYD88* p.L265P activating mutation in one patient. Several subclonal alterations, however, in the genes *SETD2*, *BRCA2*, *TERT*, and broad chromosomal regions showed heterogeneity between the brain and the eyes, between the two eyes, and among different regions of the PCNSL brain lesion. Taken together, our data show that NGS of vitreous liquid biopsies in PCNSL patients with VRL highlights shared and distinct genetic alterations that suggest a common origin for these lymphomas, but with additional site-specific alterations. Liquid biopsy of VRL accurately replicates the findings for PCNSL truncal (tumor-initiating) genomic alterations; it can also nominate precision medicine interventions and shows intra-patient heterogeneity in subclonal alterations. To the best of our knowledge, this study represents the first interrogation of genetic underpinnings of PCNSL with matched VRL samples. Our findings support continued investigation into the utility of vitreous liquid biopsy in precision diagnosis and treatment of PCNSL/VRL.

## 1. Introduction

Primary Central Nervous System Lymphoma (PCNSL) represents 4–6% of all extranodal lymphomas and is a relatively rare malignancy comprised of lymphocytes that originate in the brain [1]. They constitute 4% of all intracranial neoplasms, and most patients present in the fifth or sixth decade of life (median age of 66) with slightly higher incidence in males and Caucasians [2,3,4]. Recent epidemiological analysis has reported an incidence rate of 0.5 per 100,000 per year (~1500 new diagnoses per year) in the United States [5]. Because the neoplasm grows in the brain, signs and symptoms are initially quite subtle, making clinical suspicion an important component to obtaining an early diagnosis. Most patients present with non-specific cognitive changes as well as signs of increased intracranial pressure such as focal neurologic deficits, headache, nausea, vomiting, and seizures, as well as classic B-symptoms (fever, sweats, weight loss) [6,7]. On pathology, most PCNSL belongs to the diffuse large B-cell lymphoma subtype [8]. Treatment centers on the utilization of high-dose methotrexate, while whole-brain radiotherapy was employed in the past [9,10,11,12]. Mortality varies with other combinations of biologic agents and ranges from 3.7 to 61.9 months [13]. MRI imaging with or without biopsy (e.g., stereotactic biopsy and/or cerebrospinal fluid analysis) usually serves to make an official diagnosis.

PCNSL dissemination into other CNS tissues most commonly occurs via leptomeningeal spread into the spine, cerebrospinal fluid, testes, and importantly, the eyes [14]. Symptomatically more notable, ocular involvement is classified as vitreoretinal lymphoma (VRL), a sub-type of lymphoma within the PCNSL family [15,16]. Further, in a smaller subset of VRL cases, known as Primary Intraocular Lymphoma (PIOL), the vitreous and retina of the eye serves as the primary site, which can later lead to cerebral involvement [17]. Symptoms such as increased floaters, blurry vision, and decreased visual acuity can be mistaken for other ophthalmologic disorders (e.g., intraocular inflammation, chronic infections, metastasis, and uveal lymphoma, all of which may appear similar to VRL) [18]. Examination of the patient at the onset of such symptoms is important, especially in an individual with a known PCNSL diagnosis because 15–25% of PCNSL patients develop VRL at disease presentation and the overall 5-year survival rate of those with ocular involvement in the United States is 33.5% [15,19]. VRL liquid biopsy remains the gold-standard for confirming lymphomas, specifically sampling the vitreous and processing the sample with cytological and flow cytometry methods [20,21,22,23,24,25,26,27,28,29,30]. Depending on disease burden, ocular (vitreous/retina) tissue and cerebrospinal fluid biopsies may lack sufficient cells for successful diagnosis by cytology and flow cytometry-based methods and may result in high false-negative rates [31]. Other techniques such as DNA PCR analysis that identify B-cell immunoglobulin heavy chain (*IGH*) V(D)J gene rearrangements [32,33,34,35] and cytokine analysis with enzyme-linked immunoassay (ELISA) interleukin-6/10 levels in vitreous samples [36,37,38] complement the diagnostic workup or remain active areas of investigation. These tests, however, are indicative of clonal lymphocyte expansions—which can occur in intraocular inflammation—but do not directly assess tumor-specific features such as detection of DNA harboring signature genomic alterations.

Depending on the location of PCNSL, stereotactic needle brain biopsy may be risky, and cerebrospinal fluid analysis may be indeterminate due to a paucity of tumor cells. Thus, vitreous liquid biopsy, coupled with next-generation sequencing (NGS), can address the unmet needs above and allow for less invasive, early, definitive diagnosis and nomination of precision medicine interventions in PCNSL with VRL involvement, as commonly done in other cancer types [39,40,41]. We have previously demonstrated proof of principle for our pioneering method of interrogating undiluted and diluted small-volume vitreous biopsies in patients with VRL by NGS of cancer-associated genes [42,43]. Despite lack of matched brain lesion tissue, the aforementioned report found PCNSL genomic hallmarks in vitreous liquid biopsies such as recurrent *MYD88* mutations and *CDK2NA* gene deletions [42]. In one vitreous cytology-negative case from that report, tumor DNA was even detected two years prior to the patient presenting with a PCNSL lesion. Here, using a targeted NGS panel enriched for recurrent genetic alterations related to approved/investigational therapeutic targets, we report analysis of six samples from two patients with matched PCNSL brain tissue and vitreous liquid biopsy samples. We interrogate whether the VRL targetable genomic landscape resembles that of the corresponding PCSNL and whether PCNSL/VRL is characterized by intra-patient tumor genomic heterogeneity (ITH) between disease sites and among regions within a site in these patients. This work contributes to the establishment of molecular analysis of vitreous liquid biopsies as novel tools with potential clinical applications in enabling precision diagnosis and treatment in PCNSL/VRL patients.

## 2. Case Presentation

### 2.1. Patient Cases

Two patients with a history of PCNSL presented at the University of Michigan Health System (Michigan Medicine) with suspected VRL. The first patient (Case 1) was a 53-year-old male without history of immunodeficiency, with a prior history of non-Hodgkin lymphoma of the brain not of testicular origin, who was treated with methotrexate and rituximab and had successful remission after two years of treatment. Two years after achieving remission, the patient noted new floaters and blurred vision. Visual acuity was 20/20 in both eyes, but scant numbers of vitreous cells were present in both eyes. There were no retinal or choroidal lesions, and the retinal exam was otherwise unremarkable (Appendix A). The patient underwent sequential diagnostic and therapeutic vitrectomies of both eyes (to potentially de-bulk suspected VRL cells). Flow cytometry analysis of the vitreous humor biopsies demonstrated a kappa-restricted B-cell population comprising 48% and 23% of the small cell populations in the right (oculus dexter, OD) and left (oculus sinister, OS) eyes, respectively, thereby confirming VRL (Appendix A). A follow up MRI of the brain also demonstrated T2 FLAIR abnormalities, which was indicative of a pathological process likely due to PCNSL relapse (when associated with the flow cytometry analysis). The patient then underwent subsequent initiation of the methotrexate, rituximab, cytarabine, and temozolomide. The patient continued to receive intrathecal rituximab infusions at the time of this reporting after showing that the CSF analysis was negative for lymphoma. Three samples (one brain FFPE tissue biopsy and two vitreous liquid biopsies, one from each eye) were analyzed from this patient.

The second patient (Case 2) was a 79-year-old female without history of immunodeficiency, with enhancing brain lesions on MRI, and was diagnosed with PCNSL, which was confirmed by brain biopsy after experiencing right-sided numbness and right foot drop for one week. She also noted occasional new floaters in both eyes. Ophthalmic exam noted significantly decreased visual acuity in the left eye and a subtle decrease in visual acuity in the right eye, as well as dense vitreous veils and membranes (left worse than right) (Appendix A). She underwent a combined diagnostic and therapeutic vitrectomy of the left eye. While cytological examination of the vitreous showed a paucicellular sample, it did reveal abnormal lymphocytes, thus confirming VRL (Appendix A). She then underwent combined methotrexate, rituximab, and vincristine therapy with successful remission of her PCNSL and no evidence of VRL on subsequent ophthalmologic exams. Three samples (brain biopsy samples from two separate regions and one vitreous liquid biopsy, OS) were analyzed from this patient.

### 2.2. IGH Rearrangement Analysis of Brain and Vitreous Biopsies

The amounts of DNA obtained from the brain FFPE and vitreous liquid biopsy samples were sufficient for both rearrangement PCR and NGS analyses (Appendix A). All brain and eye samples underwent B-cell receptor immunoglobulin heavy locus (*IGH*) gene rearrangement PCR testing in duplicates. This assay has the sensitivity to detect 93% of observed *IGH* V(D)J rearrangements [44]. Since each B cell undergoes an *IGH* V(D)J rearrangement during its development, the presence of a prominent rearrangement amplicon size indicates dominance of one clone. Clonal dominance indicates clonal expansion, which is suggestive, but is alone insufficient, to support diagnosis of VRL.

Case 1 vitreous samples from each eye (1-OD and 1-OS) showed an identical predominant *IGH*-rearranged B-cell clone with rearranged PCR amplicons of identical sizes present in all three assay frameworks (1 (blue), 2 (green) and 3 (red), the colors referring to the different fluorophores used for detection of PCR amplicons; Figure 1A and Appendix A). This patient’s single brain biopsy tissue sample (1-Brain) was of lower DNA quality and was considered inadequate due to failure of the positive control band (black frame, Appendix A); however, it did show identical amplicons to its two matched vitreous samples in two of the three frameworks (2 and 3; Figure 1A and Appendix A). The samples also underwent testing for the *IGH-BCL2* t(14;18) (q32;q21) oncogenic translocation, a gene fusion found in 20–30% of DLBCLs [45,46]. Testing was carried out using a PCR assay that covers the major, intermediate, and minor fusion breakpoints with 75% overall sensitivity [44]. All three samples from this patient were negative for this translocation (Appendix A).

Case 2, despite having high quality and quantity DNA for all samples, did not display any dominant clonal *IGH* rearrangements, at least among those that are covered by this assay’s 93% sensitivity (Figure 1B and Appendix A; visible peaks in Framework 3 (red) correspond to known artifacts of this method in samples without a detectable rearrangement). These three samples were also negative for the *IGH-BCL2* gene fusion (Appendix A), which has been identified in DLBCLs and other B-cell lymphomas [47,48]. Thus, clonality and oncogenic fusion data for Case 2 were unable to provide support for the lymphoma diagnosis. Taken together, data for these two patients highlighted that while cytology was sufficient for diagnosis, flow cytometry, molecular clonality (*IGH* rearrangements), and *IGH-BCL2* oncogenic DNA fusion PCR testing did not provide definitive and consistent molecular proof of VRL across samples. However, in studies that have analyzed larger numbers of VRL patients, as we have previously reported [42], cytological analyses may be indeterminate due to paucicellular specimen (as in Case 2), and thus inclusion of more sensitive, molecular diagnostic approaches would address this unmet clinical need. Finally, the limited breadth of gene targets in these current methods does not allow for assessment of possible intra-patient tumor heterogeneity and disease evolution. This highlights the limitations of current approaches and underscores the need for high-sensitivity and specificity molecular, NGS-based liquid biopsy methods that asses the landscape of potential cancer-driving genomic alterations in VRL patients.

### 2.3. Comprehensive Genomic Analysis of Matched Brain and Vitreous Biopsy Samples

DNA NGS of brain FFPE and vitreous biopsy samples was performed using a version of the Oncomine Comprehensive Assay (OCPv1) [49], a targeted panel covering 131 oncogenes and tumor suppressor genes (TSGs) using Ion Torrent sequencing (Appendix A). OCP is one of the assays employed by the National Cancer Institute Molecular Analysis for Therapy Choice (NCI-MATCH) precision oncology clinical trial, which matches molecular alterations to targeted therapies in solid tumors, lymphomas, and myelomas [50]. All six samples passed stringent NGS quality criteria (Appendix A). Brain and vitreous samples from Case 1 harbored recurrent genomic alterations present in PCNSL and VRL. Namely, all three samples (1-Brain, 1-OD, and 1-OS) showed a deep (homozygous) deletion of *CDKN2A* (Figure 2A), an alteration being investigated as a predictive biomarker for CDK4/6 inhibitors [51]. This deletion was present at similar levels, which indicated similar tumor DNA content in the three samples (Figure 3). Likewise, a one-copy deletion of *JAK2* was also consistently present in all three samples. Other alterations that were identified instead revealed the presence of intra-patient tumor genomic heterogeneity (ITH) between the brain and vitreous samples. While sample 1-Brain harbored a low-level gain in chromosome arm 3p, it was not present in either of the two eye samples. Conversely, a broad copy gain of the distal half of chromosome arm 11q was prominent in the two eye samples but was not present in the brain biopsy sample. Likewise, a *MYD88* (p.A221T) missense mutation of unknown significance was detected only in the brain biopsy at a low, sub-clonal variant frequency of 8% indicating heterogeneity even within the brain lesion. Some ITH was even observed between the two vitreous samples: a *BRCA2* p.C3287X deleterious nonsense mutation was present only in the 1-OD sample at 36% variant frequency, indicating tumor content consistent with the proportion of the B cell population by flow cytometry (Appendix A). Close inspection of raw mutational data and NGS raw data did not show presence of these mutations in the other samples despite adequate tumor content and NGS coverage. Taken together, these data provide confirmation of PCNSL/VRL in this patient and reveal considerable sub-clonal ITH.

Case 2, a confirmed PCNSL patient by brain biopsy, had only shown a paucicellular population of tumor cells by cytology examination of the vitreous liquid biopsy and negative results for flow cytometry, *IGH* V(D)J analysis, and *IGH:BCL2* t(14;18) rearrangements in all three samples. At the genomic level, however, the two regions of the brain lesion and the single vitreous sample (2-Brain-A, 2-Brain-B, and 2-OS) showed a deep *CDKN2A* deletion (Figure 2B and Figure 4). This patient also harbored a *MYD88* (p.L265P)-activating mutation, which is an alteration that is actively being investigated as a therapeutical target [52]. Likewise, a *PBRM1* (p.R35fs) deleterious indel was detected consistently in both brain regions and the eye sample. These signature PCNSL and VRL molecular findings undisputedly confirm the lymphoma diagnosis. Furthermore, intra-patient ITH was also observed in this patient, whereby the brain samples harbored likely one-copy deletions in regions containing *FGFR1* and *TERT*, which were alterations not found in the eye. Conversely, the vitreous sample showed a frameshift indel in *SETD2*, an alteration that was exclusive to that sample. Intra-lesion ITH was also observed in the brain, whereby sample 2-Brain-B contained one-copy losses in *PTPN11* and *FLT3*, which were alterations absent in 2-Brain-A (or the eye sample). Taken together, our data continue to establish NGS as a high-sensitivity strategy to analyze a proximal biofluid (vitreous) by liquid biopsy for the diagnosis and nomination of precision medicine targets in VRL. The primary brain lesions and vitreous samples shared the predominant, probably truncal, genomic alterations but differed in the likely later-arising sub-clonal events. Our method has the potential to also facilitate the study of tumor biology by allowing identification of intra-patient and intra-tumor spatial and temporal genomic heterogeneity.

## 3. Discussion

We genomically profiled archived brain tissue biopsy samples and freshly harvested vitreous humor samples for two patients with pathologically confirmed PCNSL. There has been extensive study of extra-ocular and extra-cerebral, systemic diffuse large B-cell lymphomas (DLBCLs), which demonstrated intra-patient tumor heterogeneity within DLBCL [53,54,55]. However, due to the rarity of this type of cancer, the less common presence of ocular involvement, and the high mortality, having two patients with matched eye and brain biopsy samples presented a unique opportunity to investigate the relationship between PCNSL and VRL in distant sites. Our approach also allows sampling of an easily accessible biofluid to assess the presence of lymphoma secondary to PCNSL and may be beneficial to determine targetable alterations in cases where stereotactic needle brain biopsy is too risky or otherwise not possible due to poor underlying health of a patient with suspected PCNSL.

Our analysis suggested the imperfect sensitivity of confirming a VRL diagnosis phenotypically using current gold-standard clinical assays such as cytology, flow cytometry, and employing adjunct molecular testing like *IGH/IGK* rearrangements, and *IGH:BCL2* oncogenic fusion PCR testing on vitreous liquid biopsies. On the other hand, NGS profiling of the same liquid biopsy samples (1) overcame sensitivity issues, (2) molecularly confirmed the lymphoma diagnosis, (3) established the clonal relationship between brain and eye samples, (4) allowed detection of intra-patient subclonal tumor genomic heterogeneity between disease sites, and (5) enabled nomination of potential precision medicine interventions. This was possible due to our previously published studies demonstrating greater than 95% concordant results between our NGS method and orthogonal methods such as Sanger sequencing and quantitative PCR [49,56,57].

Thus, both cases harbored specific activating alterations in oncogenes or deleterious ones in tumor suppressors, identified as putative cancer-driving genomic events. Both cases had complete *CDKN2A* deletions, an alteration we have observed to be highly recurrent in VRL patients and which is a promising genomic biomarker with potential clinical utility that warrants further investigation [42]. *MYD88*, a gene encoding for an NF-kB pathway component, is also recurrent in B-cell lymphomas [58,59,60,61,62,63,64,65] and was found here as a truncal activating mutation in Case 2. Interestingly, Case 1 harbored a *MYD88* p.A221T mutation with a highly subclonal status in the brain tissue sample (8% variant fraction). This alteration is unreported in the literature to our knowledge, and its importance as a cancer driver is unclear. Other alterations showed heterogeneity between the brain and eye samples as well as between regions of the brain tumor and between the two eyes, consistent with a heterogeneous, evolving disease. For example, a *BRCA2* inactivating nonsense mutation (p.C3287X) in Case 1 was only observed in one eye but not in the other or the brain tumor tissue. *BRCA2* as well as *BRCA1* have been extensively studied as DNA repair genes, which, when mutated, lead to many cancer types (prominently breast cancer) but are rarely associated with B-cell lymphomas [66,67]. Importantly, complete inactivation of *BRCA2* renders tumors susceptible to PARP inhibitor targeted therapy, although we are unable to determine the zygosity status of the mutation in our sample due to its mixture with normal DNA. Single-cell analysis as has been reported by us and others [68,69], would detect zygosity in an unambiguous manner by isolating individual vitreous or CSF liquid biopsy lymphoma cells with the DEPArray™ method.

A frameshift mutation in *PBRM1* was present in all Case 2 samples, and mutations in this gene have not been previously documented in lymphoma to our knowledge. *PBRM1*, a SWI/SWF chromatin modifying protein, is inactivated in renal cell carcinoma and is an actively studied gene for therapeutic intervention [70,71]. Interestingly, a mutation in *SETD2*, another tumor suppressor gene that encodes a histone lysine methyltransferase [72], was only found in the vitreous sample, which was collected after detection of the brain lymphoma. Like *PBRM1*, *SETD2* is commonly inactivated in renal cell carcinomas as part of chromosome arm 3p loss [73].

The observation of intra-patient tumor heterogeneity in these two patients is in line with long-standing observations that almost all cancer types exhibit variations in genomic profiling, which was originally postulated by Nowell several decades ago [74]. Because these patients had recurrence of their disease, this finding is also in line with prior observations that cancer heterogeneity is a leading cause for treatment resistance and ultimately patient mortality [75]. Additionally, inter-patient tumor genomic heterogeneity was also clearly demonstrated in these two patients, similar to that observed by large-scale cancer genomics efforts such as The Cancer Genome Atlas program (TCGA) [76]. These studies provided the current understanding of cancer as having some shared driver genomic alterations between patients by cancer type or pan-cancer (e.g., the *CDKN2A* deletions in the two patients described here found by our group and others to be highly recurrent in VRL/PCNSL lymphomas [42,77,78]). However, many individual driver alterations, clonal or subclonal, are not shared by all patients, hence emphasizing the rising importance and implementation of personalized, precision oncology in treating cancer [79,80]. The mutation profile resulting from cancer genomic profiling in this work not only confirms the power of the technique, but also the implication that PCNSL and VRL likely harbor a heterogenous genomic profile at baseline.

Limitations of this study include n = 2 cases with matched PCNSL and VRL samples, which is indicative of the rarity of this cancer type that involves the brain and both eyes. This limits the generalizability of our genomic findings. Larger cohorts of matched VRL liquid biopsies and tumor tissue will need to be analyzed to establish the sensitivity and specificity of the liquid biopsy approach with respect to tumor tissue. Further, our targeted 131-gene panel may not capture the entire landscape of alterations in each patient.

## 4. Materials and Methods

### 4.1. Case Selection

This study was carried out with the highest ethical standards and with the approval of the University of Michigan Institutional Review Board (Protocol # HUM00085419). We selected patients that had archived, flash-frozen (samples immediately placed in a −20 °C environment after collection followed by storage at −80 °C) vitreous specimens that were clinically suspected to have VRL and known to have tissue biopsy-proven PCNSL (either prior or at time of vitreous biopsy). All samples were linked with clinicopathological information from the clinical archive. Hematoxylin and eosin (H&E) stained slides and Immunohistochemical (IHC) stains (where available) were reviewed by a board-certified neuropathologist to ensure PCNSL presence and sufficient tumor content.

### 4.2. Sample Processing and IGH Rearrangement PCR Testing

For vitreous liquid biopsy samples, we used first-wash and/or diluted vitreous fluid obtained at vitrectomy as the pure vitreous sample, and early washes had presumably greater concentrations of any cells present compared to subsequent washes. Vitreous was obtained using a 25-gauge vitrector attached to a 3cc syringe. The syringe was emptied of its vitreous contents into an Eppendorf tube that was immediately placed in a −20 °C environment. Long-term storage of the sample utilized a freezer set at −80 °C. Once ready to process, the sample was thawed on ice and centrifuged at 2000× *g*, then 4000× *g* for 5 min each time. Genomic DNA was isolated from the cell pellets using the Qiagen AllPrep FFPE DNA/RNA kit (QIAGEN, Hilden, Germany) with the following modifications for non-formalin-fixed, paraffin-embedded (non-FFPE) samples: 1. No de-paraffinization treatment. 2. First 56 °C incubation reduced to 1 min. 3. Incubation at 90 °C was omitted. DNA samples were quantitated with the Qubit Fluorometer (ThermoFisher, Waltham, MA, USA).

Brain tumor tissue DNA was isolated as previously described [81]. Briefly, for each brain lesion tissue sample, 3–5 mm × 10 μm FFPE sections were cut from one and two FFPE blocks for case 1 and 2, respectively, enabling two-region sequencing of the brain lesion in case 2. H&E slides were examined by a neuropathologist and high tumor content areas were identified. Macrodissection of these areas with a scalpel under the microscope was performed to enrich tumor content. DNA was isolated using the Qiagen Allprep FFPE DNA/RNA kit (Qiagen) and quantified with the Qubit Fluorometer (ThermoFisher).

The DNA was used for PCR testing of B-cell receptor *IGH* gene V(D)J rearrangements and *IGH-BCL2* oncogenic translocations at the University of Michigan Clinical Molecular Diagnostics lab. The *IGH* rearrangement assay uses BIOMED-2 primers which are capable of detecting clonal populations in ~90% of B-cell lymphoid malignancies [44]. The *IGH-BCL2* assay detects fusions involving the major (mbr), minor (mcr), and intermediate (icr) breakpoint cluster regions, which collectively account for ~75% of *IGH-BCL2* fusions.

### 4.3. Targeted Next-Generation Sequencing (NGS)

Targeted NGS was performed as previously described [49,57,81,82]. Briefly, 20 ng DNA isolated above underwent Ion Torrent NGS (ThermoFisher) using the DNA component of a beta version of the Oncomine Comprehensive Assay (OCPv1) [49], a custom panel comprised of 3434 amplicons targeting 131 oncogenes and tumor suppressor genes (TSGs, Appendix A). The panel was constructed by selecting recurrent genomic alterations from large-scale NGS and copy-number microarray data sets of pan-solid tumor and lymphoma samples. Selected targets cover recurrent somatic mutations, indels and copy changes in oncogenes, and tumor suppressor genes with an emphasis on approved/investigational therapeutic targets [49].

Library preparation with barcode incorporation, template preparation, and sequencing using the Ion Torrent Proton sequencer were performed according to the manufacturer’s instructions with PCR cycle number adjustment for input DNA amounts above or below the recommendation (3 cycles per ~10-fold difference in DNA amount). Data analysis was performed using Torrent Suite 4.0.2, with alignment by TMAP using default parameters, and variant calling using the Torrent Variant Caller plugin (version 4.0-r76860) with default low-stringency somatic variant settings.

### 4.4. NGS Mutation and Copy Number Analysis

Variant annotation filtering and prioritization was performed essentially as described using validated in-house pipelines [49,56,57,83]. Briefly, called variants were filtered to remove: (1) synonymous or non-coding variants, (2) those with flow-corrected coverage read depths (FDP) less than 20, (3) flow-corrected variant allele contained reads (FAO) less than 6, (4) variant fractions (FAO/FDP) less than 0.05, (5) extreme skewing of forward/reverse flow-corrected reads calling the variant (FSAF vs. FSAR, >5-fold difference), and (6) indels within homopolymer nucleotide ran > 4. Called variants were then filtered using a panel-specific, in-house blacklist. Remaining variants with population allele frequencies >0.5% in EXAC or the 1000 Genomes project (KG), and those reported at any population allele frequency in EXAC or 1000 genomes and with observed variant fractions between 0.40 and 0.60 or >0.9 (likely heterozygous and homozygous, respectively), were considered germline variants unless occurring at a known hot spot. High confidence somatic variants passing the above criteria were then visually confirmed in Integrative Genomics Viewer (IGV, available online: https://www.broadinstitute.org/igv/ accessed on 15 April 2021). The excluded variants comprised those located at the last mapped base (or outside) of amplicon target regions, variants with the majority of supporting reads harboring additional mismatches or indels (likely sequencing errors), those in repeat-rich regions (likely mapping artifacts), and those occurring exclusively in one amplicon if overlapping amplicons cover the variant. Our group has previously confirmed that these filtering criteria identify prioritized high-confidence somatic variants that pass Sanger sequencing validation with >95% accuracy [46,57,84,85,86].

Copy number (CN) analysis from total amplicon read counts provided by the Coverage Analysis Plug-in (v4.0-r77897) was performed essentially as described using a validated approach [49,56,57,81,87]. Briefly, amplicon-level CN was determined as the log2 (CN ratio) between read count in the tumor sample and that in a composite of non-patient-matched normal samples, normalized for sequencing depth and GC nucleotide content. Gene-level CN estimates were then calculated as coverage-weighted averages of amplicon-level log2 (CN ratios). Genes with a log2 CN ratio estimate of <−1 or >0.6 were considered to have high-level losses or gains, respectively.

We then prioritized as putative cancer-driving alterations those that were (1) deleterious in TSGs (nonsense, frameshift, homozygous deletion), (2) recurrent “hotspot” mutations (in the “Curated set of non-redundant studies” cohort in cbioportal.org or those with solid literature evidence) in oncogenes or TSGs, (3) those with OncoKB cancer-driver annotation in oncogenes or TSGs, or (4) high-level copy number amplifications in oncogenes. Variants without support from any of those categories were designated as variants of unknown significance (VUS).

## 5. Conclusions

To the best of our knowledge, this work represents the first effort to genomically compare PCNSL and VRL in individuals harboring both lesions. These data provide foundational insights to better understand the genetic relationship between these two related yet site-distinct tumors and to devise strategies toward molecularly informed, precision diagnostic and therapeutic decision-making for this difficult-to-diagnose and deadly disease.

## Figures and Tables

**Figure 1 ijms-22-09992-f001:**
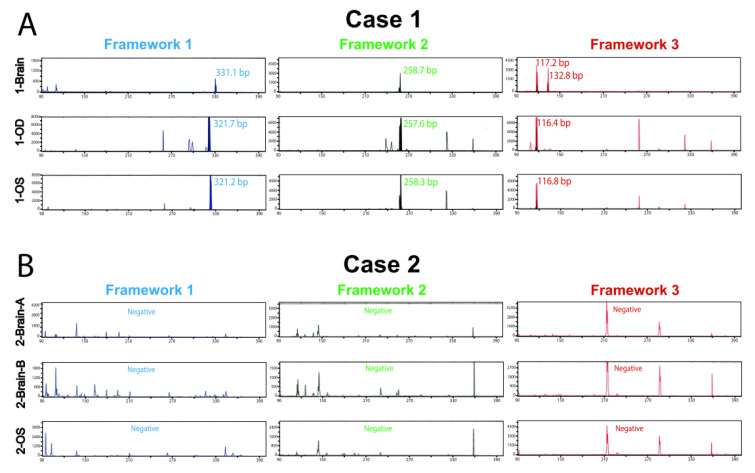
Clonality testing in brain and vitreous liquid biopsy lymphoma samples. DNA obtained from brain PCNSL FFPE tissue and vitreous fluid underwent clonal *IGH* rearrangement PCR testing covering ~90% of the most common rearrangements. Assays were run in duplicate with negative controls (one replicate shown per sample). Assay includes three rearrangement frameworks, 1 (blue), 2 (green), and 3 (red), the colors indicating different fluorophores used to detect the PCR amplicon. Samples containing clonal populations can show rearrangement product (peaks) in one or more frameworks. Clonal rearrangements were defined as peaks identical between replicates that were at least 2× higher than the third highest peak. Capillary electropherogram plots of band intensity over fragment length show a positive PCR result for *IGH* gene rearrangement in Case 1 (**A**) but not Case 2 (**B**).

**Figure 2 ijms-22-09992-f002:**
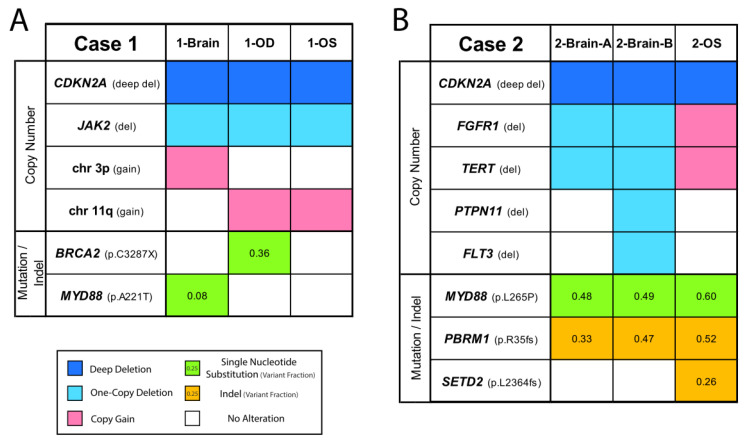
Genomic analysis of matched brain tissue and vitreous liquid biopsy samples for Cases 1 and 2. Integrative heatmaps showing genomic alterations detected in Case 1 (**A**) and Case 2 (**B**). NGS using the OCPv1 panel was performed on each DNA sample. Gene copy number losses (navy blue for deep and light blue for shallow deletions) and gains (red), as well as gene mutations (green) and short insertions/deletions (indels, brown) with the respective amino acid changes are shown. Numbers inside mutation or indel boxes represent the variant fraction, i.e., fraction of mutated sequencing reads with respect to total reads covering that base. Variant fractions deviate from 0.5 and 1.0 due to normal DNA mixed with the tumor DNA sample. Unfilled boxes represent wild-type status for that position.

**Figure 3 ijms-22-09992-f003:**
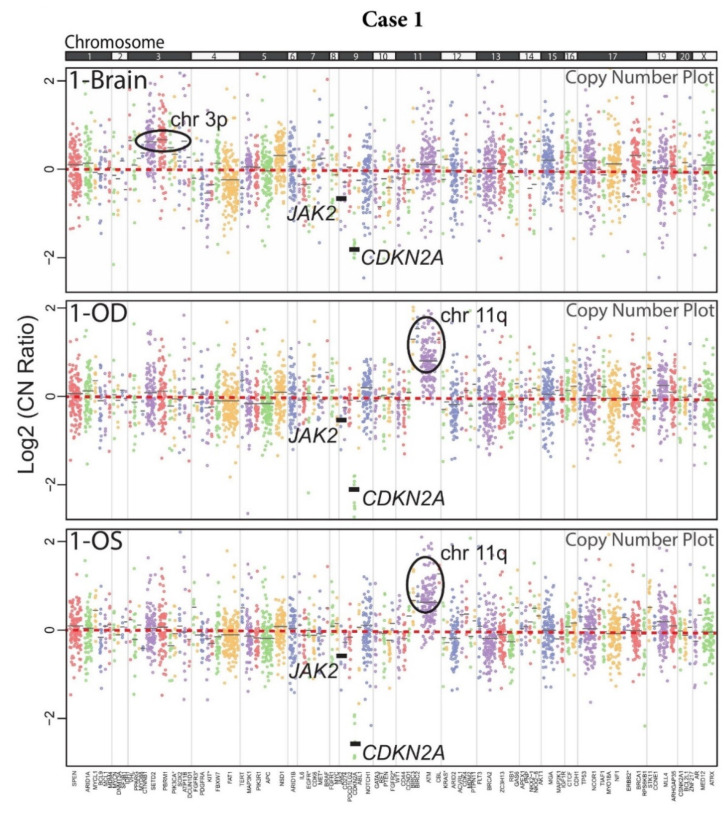
Copy number plots of matched brain tissue and vitreous liquid biopsy samples for Case 1. Copy number plots for each sample are shown as log2 copy number ratio (amplicon-level ratio between read counts in the tumor sample and read counts in a composite of normal samples, normalized for sequencing depth and GC content). Dots represent individual amplicon log 2 copy number ratios. Dots of the same color represent a gene, and black horizontal bars represent average gene-level log2 copy number ratio estimates (coverage-weighted). Genes are listed at the bottom in chromosome order, and chromosomes are separated by gray lines and marked at the top. Genes marked with * have two groups of amplicons with different colors covering them. Altered/relevant genes are highlighted with thick, black gene-level log2 copy number ratio lines. Altered, broad genomic regions are marked with black ovals.

**Figure 4 ijms-22-09992-f004:**
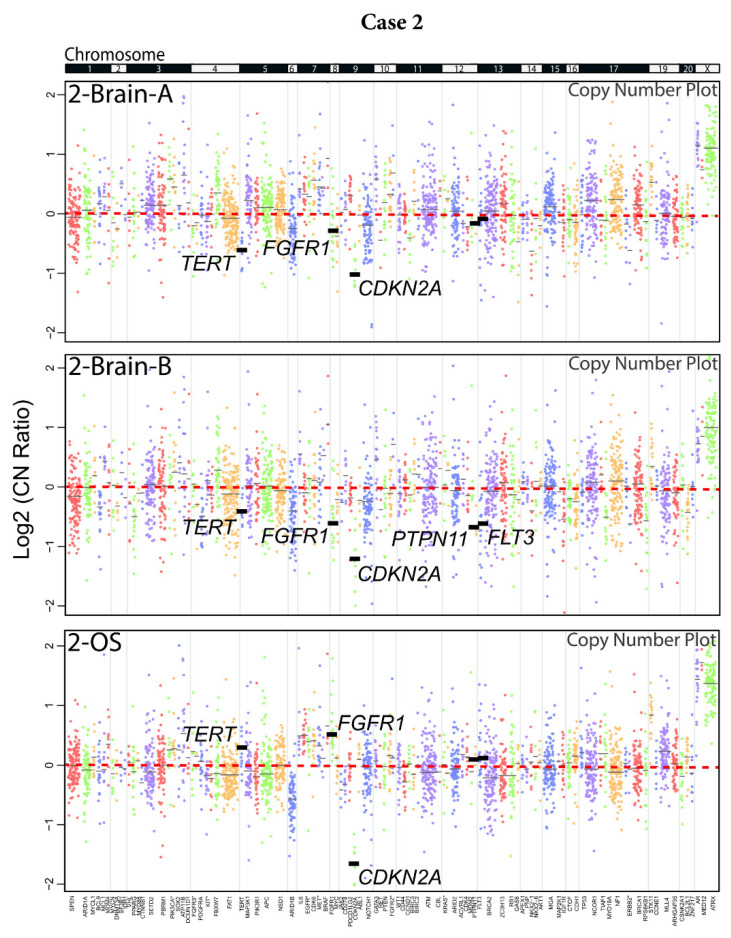
Copy number plots of matched brain tissue and vitreous liquid biopsy samples for Case 2. Copy number plots for each sample are shown as log2 copy number ratio (amplicon-level ratio between read counts in the tumor sample and read counts in a composite of normal samples, normalized for sequencing depth and GC content). Dots represent individual amplicon log 2 copy number ratios. Dots of the same color represent a gene, and black horizontal bars represent average gene-level log2 copy number ratio estimates (coverage-weighted). Genes are listed at the bottom in chromosome order and chromosomes are separated by gray lines and marked at the top. Genes marked with * have two groups of amplicons with different colors covering them. Altered/relevant genes are highlighted with thick, black lines for gene-level log2 copy number ratios.

## Data Availability

Data available upon request.

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
