# Peer review of "Comparative Molecular Analysis of Primary Central Nervous System Lymphomas and Matched Vitreoretinal Lymphomas by Vitreous Liquid Biopsy"

_ijms, 2021, doi:10.3390/ijms22189992_

Round 1
Reviewer 1 Report
This manuscript, original research, written by Dr. Daniel A. Balikov et al., with the title of “Comparative Molecular Analysis of Primary Central Nervous System Lymphomas and Matched Vitreoretinal Lymphomas by Vitreous Liquid Biopsy” describes two cases of primary central nervous system lymphoma (PCNSL) with vitreoretinal lymphoma (VRL).
Primary central nervous system lymphoma (PCNSL) is an uncommon variant of extranodal non-Hodgkin lymphoma (NHL) that involves the brain, leptomeninges, eyes, or spinal cord without evidence of systemic disease. The most common histopathological subtype of PCNSL is diffuse large B-cell lymphoma (DLBCL). This article is well written, it is easy to read, have enough tables, figure, and references. Before publishing this research, the authors could address the following minor comments.
Minor comments:
1- The most notable risk factor for developing PCNSL is immunodeficiency, and this association may contribute to the pathogenesis of disease. Did the patients have any condition associated with immune deficiency (for example, HIV, immune suppressant treatment, etc)?
2- Patient 1 had a previous history of MTX treatment, which associates to lymphoproliferative disease. May the authors comment about MTX and lymphomagenesis?
3- Testicular lymphoma has an increased propensity for CNS involvement, especially in the parenchymal compartment. Had the patient 1 evaluation of the testicles by examination and ultrasound?
4- Page 10 of 15, line 337-8: “This study was carried out with the highest ethical standards and with the approval of the University of Michigan Institutional Review Board.” Could you please provide the IRB reference number/letters (if available)?
5- Page 10 of 15, line 339: “flash-frozen vitreous specimens”. Could you please explain what you mean by “flash-frozen”?
6- Page 10 of 15, lines 339-340: “specimens that were clinically suspected to have VRL and known to have tissue biopsy-proven PCNSL (prior, or at time of vitreous biopsy).” Could you please specify if the cases were primary vitreoretinal lymphoma (VRL), or Primary Central Nervous System Lymphoma (PCNSL)? Or at this stage of the research this fact was unclear yet?
7- Regarding Figure 1. The authors could explain in more detail the meaning of blue, green, and red frames because not all readers may be familiar with the structure or nomenclature (V-gene, green; D-gene, blue; J-gene,?).
8- Is there any information about the morphology and immunophenotypes?
9- Since MDPI is flexible about the number of tables and figures, the authors could add the supplementary information in the main text so the readers won’t need to download a supplementary file.
Author Response
This manuscript, original research, written by Dr. Daniel A. Balikov et al., with the title of “Comparative Molecular Analysis of Primary Central Nervous System Lymphomas and Matched Vitreoretinal Lymphomas by Vitreous Liquid Biopsy” describes two cases of primary central nervous system lymphoma (PCNSL) with vitreoretinal lymphoma (VRL).
Primary central nervous system lymphoma (PCNSL) is an uncommon variant of extranodal non-Hodgkin lymphoma (NHL) that involves the brain, leptomeninges, eyes, or spinal cord without evidence of systemic disease. The most common histopathological subtype of PCNSL is diffuse large B-cell lymphoma (DLBCL). This article is well written, it is easy to read, have enough tables, figure, and references. Before publishing this research, the authors could address the following minor comments.
We thank the reviewer for the compliments.
Minor comments:
1- The most notable risk factor for developing PCNSL is immunodeficiency, and this association may contribute to the pathogenesis of disease. Did the patients have any condition associated with immune deficiency (for example, HIV, immune suppressant treatment, etc)?
The reviewer brings up an interesting point. Upon careful re-review of the patient charts, neither patient had any condition associated with immune deficiency. We have added a phrase in each patient’s description to address this point.
2- Patient 1 had a previous history of MTX treatment, which associates to lymphoproliferative disease. May the authors comment about MTX and lymphomagenesis?
The reviewer raises an interesting and important point. There is evidence in the literature of MTX causing lymphomas (examples include PMID: 8849387, 31471410, 33483528). In the case of patient 1, there was no prior use of MTX before the first diagnosis of PCNSL. Furthermore, at the time of recurrence, MRI imaging demonstrated localization of lymphoma only in the eyes and the brain, and no other extranodal site. MTX-associated lymphoma predominantly presents with multiple extranodal sites and the lack of lymph node involvement in this patient case decreased the likelihood that the prior MTX treatment gave rise to the recurrence of disease.
3- Testicular lymphoma has an increased propensity for CNS involvement, especially in the parenchymal compartment. Had the patient 1 evaluation of the testicles by examination and ultrasound?
The reviewer brings up an interesting point. Upon re-review of the medical records, the patient did have a testicular exam, which did not document any abnormal masses, but there was no documentation of a testicular ultrasound. We have added a phrase in the patient’s description to address this point.
4- Page 10 of 15, line 337-8: “This study was carried out with the highest ethical standards and with the approval of the University of Michigan Institutional Review Board.” Could you please provide the IRB reference number/letters (if available)?
We have provided the following in the methods section: Genetic and Epigenetic Analysis of Tissues from Patients with Eye and Orbital Disease (HUM00085419). Approved 9/17/2014 - 2/10/2022
5- Page 10 of 15, line 339: “flash-frozen vitreous specimens”. Could you please explain what you mean by “flash-frozen”?
When the vitreous biopsy is obtained via a vitrector attached to a 3 cc syringe, the vitrector is removed and the syringe contents are emptied into a sterile Eppendorf tube. This tube is then closed, immediately placed in a -20C environment and then transferred to a -80C freezer. We have provided further explanation of that in the Methods section.
6- Page 10 of 15, lines 339-340: “specimens that were clinically suspected to have VRL and known to have tissue biopsy-proven PCNSL (prior, or at time of vitreous biopsy).” Could you please specify if the cases were primary vitreoretinal lymphoma (VRL), or Primary Central Nervous System Lymphoma (PCNSL)? Or at this stage of the research this fact was unclear yet?
We thank the reviewer to pointing out the need for this clarification. The two patients were known to have PCNSL: patient 1 presented to us with recurrence that first manifested as VRL which subsequently showed imaging findings of PCNSL; patient 2 had a new brain lesion that was identified as PCNSL but with subtle changes in visual acuity. At the time of obtaining samples, specific staging was not finalized as the vitrectomies were performed to help with this for the oncologists.
7- Regarding Figure 1. The authors could explain in more detail the meaning of blue, green, and red frames because not all readers may be familiar with the structure or nomenclature (V-gene, green; D-gene, blue; J-gene,?).
We thank the reviewer for pointing out this detail. We have amended the text and figures to state that the blue, green, and red colors merely indicate different fluorophores used to detect three different subsets (frameworks) of the entire set of PCR amplicons that target possible V(D)J rearrangements (a total of three separate frameworks to detect a clonal population). They do not correspond to V, D, or J genes. If any, or more than one frameworks, demonstrate similar clonal peaks, the sample is considered a clonal population.
8- Is there any information about the morphology and immunophenotypes?
We are assuming the reviewer is referring to the pathology and flow cytometry data. Unfortunately, we have no other recorded information from the pathologists other than what is documented in the patient medical records, which were reviewed for this manuscript.
9- Since MDPI is flexible about the number of tables and figures, the authors could add the supplementary information in the main text so the readers won’t need to download a supplementary file.
We appreciate this thought from the reviewer. We decided to keep supplemental data in the supplementary file as to not bog down the reader with too many figures that demonstrate the efficacy of our methods. If the editors also agree with putting supplementary materials into the main text, we would be more than happy to do so. Editors, please advise.
Reviewer 2 Report
This article reports on targeted DNA next generation sequencing (NGS) of matched brain and vitreous samples in two patients, who each harbored vitreoretinal lymphoma (VRL) and Primary Central Nervous System Lymphoma (PCNSL). They showed a clonal relationship between the eye and brain samples in both patients which carried clonal CDKN2A deep deletions, a highly recurrent alteration in VRL patients, as well as MYD88 p.L265P activating mutation in one patient. The major concern is 2 cases findings limit the significance of the utility of vitreous liquid biopsy by NGS in precision diagnosis. Minor concern is that the authors should describe more about how to get definite vitreous tumor cells during vitrectomy.
Author Response
This article reports on targeted DNA next generation sequencing (NGS) of matched brain and vitreous samples in two patients, who each harbored vitreoretinal lymphoma (VRL) and Primary Central Nervous System Lymphoma (PCNSL). They showed a clonal relationship between the eye and brain samples in both patients which carried clonal CDKN2A deep deletions, a highly recurrent alteration in VRL patients, as well as MYD88 p.L265P activating mutation in one patient. The major concern is 2 cases findings limit the significance of the utility of vitreous liquid biopsy by NGS in precision diagnosis. Minor concern is that the authors should describe more about how to get definite vitreous tumor cells during vitrectomy.
We thank the reviewer for the helpful comments and concerns.
With respect to the significance of the utility of a vitreous liquid biopsy analysis by NGS for precision diagnosis, with these two cases of a relatively rare tumor/presentation, we hoped to illustrate the potential utility of showing how understanding the underlying mutations detected in the PCNSL and VRL samples relate to each other in a diagnostic setting. Additionally, by demonstrating the relationships between the two, it will hopefully encourage further work into developing/employing precision treatments that target both the PCNSL and VRL simultaneously. We agree that our sample size is modest and accumulating a substantial set of such samples remains our priority. We have re-emphasized this point in the discussion of our limitations in the manuscript. We felt that the report on these two cases we had so far would be of interest to the PCNLS/VRL clinical and research community and would demonstrate feasibility and proof of concept.
With respect to obtaining the tumor cells during vitrectomy, we added language in the methods clarifying that a cell pellet is obtained after centrifuging the vitreous biopsy. Vitreous is an acellular material and any presence of cells on examination is abnormal. Hence, any cells obtained from the pellet are cells of interest, including potentially vitreous tumor cells. Another contributing factor is using the original vitreous fluid or subsequent washes as the specimen of choice. The pure vitreous and early washes (in that order) are preferable as they will have a greater concentration of any cells present compared to later (increasingly dilute) washes. Samples used here for DNA analysis were obtained from first and subsequent washes. We have emphasized that point in the manuscript.
Reviewer 3 Report
General comments:
In this paper, Balikov et al report on targeted DNA next generation sequencing of matched brain and vitreous samples in two patients, who each harbored primary central nervous system lymphoma and vitreoretinal lymphoma. The findings are interesting and informative. The authors observed a clonal relationship between the eye and brain samples in both patients which carried clonal CDKN2A deep deletions. The findings also show the heterogeneity between the brain and the eyes, between the two eyes, and among different regions of the primary central nervous system lymphoma lesion. These findings observed in these rare cases are informative for the readers.
Author Response
General comments:
In this paper, Balikov et al report on targeted DNA next generation sequencing of matched brain and vitreous samples in two patients, who each harbored primary central nervous system lymphoma and vitreoretinal lymphoma. The findings are interesting and informative. The authors observed a clonal relationship between the eye and brain samples in both patients which carried clonal CDKN2A deep deletions. The findings also show the heterogeneity between the brain and the eyes, between the two eyes, and among different regions of the primary central nervous system lymphoma lesion. These findings observed in these rare cases are informative for the readers.
We thank the reviewer for their time, expert reviewing of our manuscripts and for theircompliments. We agree that these published findings will hopefully help both the clinical and scientific community in studying and treating this terrible disease. We strive to accumulate a substantial set of such samples and improve our methods to enable more comprehensive molecular characterization of this malignancy with the ultimate goal of translating our findings to impact patient care.
Round 2
Reviewer 2 Report
1.It seems difficult to explain the inconsistent results from NGS analysis in figure 2. For example, in case 1, why the BRCA2 and MYD88 mutation are different from 1-OD and 1-OS. In case 2, why the copy number of PTPN11 and FLT3 are not consistent between Brain-A and Brain-B. Besides, More patient sample counts need to be verified. The profile of genes mutation from 2 patients are almost different except for CDKN2A gene deletion. 2.It might need different approach to double-check the results of genes mutation by NGS. For example, the analysis of DNA copy number might be further confirmed by a real-time PCR.
Author Response
We thank the reviewer for reviewing our first revision submission for our manuscript. Please find our response to their new comments below.
REVIEWER 2
1.It seems difficult to explain the inconsistent results from NGS analysis in figure 2. For example, in case 1, why the BRCA2 and MYD88 mutation are different from 1-OD and 1-OS. In case 2, why the copy number of PTPN11 and FLT3 are not consistent between Brain-A and Brain-B. Besides, More patient sample counts need to be verified. The profile of genes mutation from 2 patients are almost different except for CDKN2A gene deletion. 2.It might need different approach to double-check the results of genes mutation by NGS. For example, the analysis of DNA copy number might be further confirmed by a real-time PCR.
We thank the reviewer for providing the specific cases and mutations that appear confounding. We aimed to highlight both in the abstract and main text that these patient samples demonstrated both inter-patient and intra-patient cancer heterogeneity. As the reviewer may appreciate, intra-patient tumor heterogeneity was postulated a few decades ago by Peter Nowell (PMID: 959840). The cancer genomics era, made possible by advances in next generation sequencing technology such as the one we used in this project, revealed genomic data that supported that prescient postulate, beginning with the report by Gerlinger et al (PMID: 22397650) on multi-region sequencing of kidney tumors. A wealth of data following that report has shown this to be the case in almost all cancer types and almost all individual patients studied. This has wide ranging implications in diagnosis, treatment and outcomes. That is because cancer heterogeneity is thought to be a major cause of development of treatment resistance and ultimately mortality (an idea originally brought forth initially by another early postulate by Fidler et al. (PMID: 6985715). In fact, while cancers are known to acquire de novo mutations that result in treatment-refractory disease spread, there are also cancers that contain heterogenous subclones, that manifest as initially being treatment responsive but then become resistant.
Additionally, inter-patient tumor genomic heterogeneity was also clearly demonstrated by large scale cancer genomics efforts such as the TCGA (PMID: 29625048). These studies provided the current understanding of cancer as having some shared driver genomic alterations between patients by cancer type or pan-cancer (e.g. the CDKN2A deletions in our two patients found by our group and others to be highly recurrent in VRL/PCNSL lymphomas (PMID: 28002793, PMID: 25991819, PMID: 34448823). However, many individual driver alterations are found, clonal or subclonal, that are not found in all patients. Each patient can be thought of as having a constellation of driver genomic alterations in their tumor that is rather specific to that patient. This, as the reviewer may know, has given rise to the field of personalized, or precision oncology, i.e. finding and targeting tumor alterations in an individualized manner for each patient (PMID: 25023344, PMID: 30603737).
For our two patients reported in the enclosed manuscript, we were not surprised to find that the genomic make up adopted these two qualities: inter- and intra-patient tumor heterogeneity. The mutation profile resulting from NGS not only confirms the power of the technique but also the implication that PCNSL and VRL likely harbor a heterogenous genomic profile at baseline (although we acknowledge we only have two cases in this report). We hoped that these findings could drive the point that these cancers can be very complex and have significant implications on how they may be treated with targeted therapy. We appreciate the reviewer’s concern on these points and we have revised the manuscript to emphasize this point accompanied by several references mentioned above.
With respect to validating the NGS findings with RT-PCR, we, unfortunately, no longer have remaining DNA (or RNA) for undergoing such confirmation studies as it was isolated in small amounts and used for NGS. However, we would like to highlight that the method used for the enclosed study was previously confirmed by us to be greater than 95% concordant with orthogonal methods, namely Sanger sequencing and qPCR (PMID: 25593300, PMID: 25502898, PMID: 25925381). We have revised the manuscript to emphasize this point and provided these references. We hope the reviewer will find that solution acceptable.
Round 3
Reviewer 2 Report
Accept